# Thematic Patterns of Disinformation about COVID-19: The Framing of Checks in the Fato ou Fake and Lupa Agencies

**Juliana Teixeira** [1,*] **and Allysson Martins** [2,*]

1 Department of Social Communication, Federal University of Piauí, Teresina 64049-550, Brazil
2 Department of Journalism, Federal University of Rondônia, Porto Velho 76801-059, Brazil
* Correspondence: teixeira.juliana.rj@gmail.com (J.T.); allyssonviana@unir.br (A.M.)

**Abstract:** This article, which integrates broader research, aims to identify the fake news patterns propagated in the process of disinformation about COVID-19 that were evaluated by the Brazilian fact-checking agencies Fato or Fake and Lupa. Aiming at this goal, we considered the strategies for spreading false information about the disease from January to September 2020. As a methodology, we used part of the procedures associated with media framing, focusing on the themes and labels of the checked information. Politics and death were the two main issues in misinformation assessed by the agencies, closely followed by themes related to cure and prevention. Personalities were particularly relevant at Lupa. The high frequency of the political issue reveals the ideological polarization that Brazil is experiencing, leading to global health crises such as the new coronavirus pandemic.

**Keywords:** fact checking; disinformation; COVID-19

## 1. Introduction

Disinformation is a problem that was scientifically investigated in the early 2010s, especially because it jeopardizes democracy when it spreads false information, which is easily disseminated and difficult to correct (Nyhan and Reifler 2012). The context of cyber-culture (Lévy 1999; Santaella 2003; Lemos 2007; Jenkins 2009) allowed greater production and dissemination of content for those who have access to digital technologies, especially based on the logic of the connection culture (Jenkins 2014). This type of democratization of information reduces the exclusivity that media companies had in the communication process, especially with its circulation in society, at a time when the journalistic institution is increasingly discredited.

It is practically a consensus that fake news, with its disinformation process, proliferates when newspapers diminish their credibility and their exclusivity as components of a reference institution for the production and circulation of information (Eichler et al. 2018; Silva and Melo 2020; Spinelli and Santos 2018). The producers and propagators of fake news, even so, rely on the productions of this institution so that they acquire more credibility or, at least, reduce that of the media professionals. To ensure that the lie is trustworthy, the contents are in line with the beliefs and values of most consumers (Gruszynski et al. 2020; Teixeira and Costa 2020; Roxo and Melo 2018), in addition, the dissemination of fake news often comes from trustworthy people, such as friends and relatives, and it reaches only those who they agree with, given the filters of social networks. These materials also seek to replicate testimonies, legitimizing themselves through a first-person perspective, even if fictitious, digitally edited, or manipulated for decontextualization.

The spread of fake news, when it involves health-related issues, such as the anti-vaccination movement and miraculous treatments, also leads to death, such as in the yellow fever outbreak in Brazil, when more than 130 monkeys died in less than a month, in the state of Rio de Janeiro. This spread of lying is amplified by the discrediting of institutions other than journalism, such as science and politics/the state (Gruszynski et al. 2020; Teixeira

and Costa 2020). Specifically about false information about health, it is worth noting that they spread through fear and concern, even though they promise the maintenance of health, disregarding that individual actions can collectively impact society.

In this current context of disinformation, fact-checking initiatives have been seen as one of the solutions, mainly because it is a growing phenomenon in Brazil and in the world. This research does not focus on the impact of this false information on interactors but on the production process undertaken to combat it by the checking agencies, especially with regard to the themes checked, as the impacts of misinformation go beyond confusion between what is true or false. Fake news can change people's behavior, culminating even in physical violence, as in the case of the shooter who broke into a pizzeria because he believed it was home to a pedophilia network, a lie propagated after the owner supported Hillary Clinton in the 2016 presidential campaign. The purpose of this article, which is part of a broader survey, seeks to identify the patterns of fake news propagated in the process of disinformation about COVID-19 that were evaluated by the Brazilian fact-checking agencies Fato or Fake and Lupa.

It takes into consideration the reality interpretation schemes in the properties and operations of spreading false information about the new coronavirus in the period of January to September 2020. As a methodology, for this article, we used part of the procedures associated with the analysis of the media framework proposed by Entman (1993), Vimieiro and Dantas (2009) and Vimieiro and Maia (2011), focusing especially on the themes and labels of the information checked.

## 2. Framing Fake News Checks

The common imprecision of the term fake news is due, among other factors, to the multiplicity of aspects included in their studies, related to six dimensions: satire—humor based on facts; parody—humor with news creation; manufacturing—invention of information or manipulation; editing or decontextualizing media formats, facilitated by the digital aspect; advertising—paid content in journalistic format; and advertising—people and institutions paid to guide public opinion without transparency (Tandoc et al. 2018). Although the inaccuracy common in the first studies in the area persists (prior to its popularization), it can be reduced to three notions: (1) unintentional errors, leading to the publication of false information even in journalism; (2) parody of journalism, with satires bringing false (or exaggeratedly true) information in the journalistic structure and format; and (3) deliberate lies, with intentional production to spread false information, with little observance in journalism.

The first two cases are still observed during the last century, with the idea of fake news appearing for the first time in 1925 in Harper Magazine to address the mistakes of press professionals and the sensationalism that prevailed in the US press in the 19th century (Schudson 2008; Schudson and Zelizer 2017). Even so, in journalism, "the objective will never be to deceive readers, which leads us to want to separate false news (journalism) from fake news (manipulation and misinformation)" (Meneses 2018, p. 42). At the end of the 20th century, studies began to show humorous productions that parodied journalism, something that exists even today. The intention here is not deception either, but entertainment, as in the Brazilian television cases of *Casseta & Planeta, Urgente!, Furo MTV* and *Fora de Hora*, in addition to the sites The Piauí Herald, Sensacionalista, Diário de Barrelas, O Bairrista, G17 and Diário Pernambucano.

The three dimensions comprise a heterogeneous ecosystem of false information, although only the last one presents purposes, in fact, of disinformation. As they are completely different dimensions, with divergent rules and meanings, the current use of the expression has increasingly distanced it from the first two occasions, which existed before the popularization of the internet and are sometimes called false news. While "fake" refers to fraud (disinformation), "false" would be closer to incorrect (misinformation).

Checking agencies were not created along with the fake news phenomenon but to assess the exaggerations of campaigning politicians, which are not necessarily lies. "Fact-

checking departments emerged in American magazines in the 1920s and 1930s" (Santos 2019, p. 61); however, unlike what happens today, the lies were not emphasized. With the popularization of fake news, especially as of 2016, in addition to the statements made by politicians, the agencies began to emphasize misinformation in the widest and most diverse of themes.

These corrections, which would effectively allow for rectifying wrong perceptions, do not always have an effect, especially when they come from journalists, professionals who have become little credible for those who believe in the broadcasted lie—or at least do not question it, especially when they reinforce their convictions and its ideals. The effectiveness of these checks also varies according to the ideological pole of the corrected information consumer, both in the USA (Nyhan and Reifler 2012) and in Brazil (Lelo 2019). In addition to people who believe in fake news, there are those who know it is false information and yet share it as if it were true.

From checkers of politicians, agencies broaden their focus and work with statements from all kinds of public figures—published by press offices or by press media—and with fraudulent content that circulates on social networks and in society. Lupa, for example, adopts this debunking function more strongly from 2018 onwards. According to Dourado (2019), until then, Brazilian fact-checking agencies, in general, focused on the statements of politicians, in their own or traditional media, rather than in the circulation of fake news. Once they have emerged to evaluate the speech of politicians and public figures, these agencies currently appear as a "vaccine" against fake news and join other combat strategies, such as the exclusion of content and demonetization of propagators by companies such as Facebook, Google and Instagram, in addition to the enactment of laws aimed at punishing the creators and circulators of these lies (Eichler et al. 2018; Gruszynski et al. 2020; Silva and Melo 2020; Spinelli and Santos 2018).

The main presentation of the check tends to bring labels, with its content usually having links which allow to verify images, infographics and videos in order to facilitate the understanding of consumers (Nyhan and Reifler 2012; Fernandes et al. 2019). Labels have a common use in checking agencies, although their choices are subjective and create distortions even about what would or would not be a lie (and the gradations included). Although agencies hardly ever check the same statements, as there is a certain understanding that the job is to assess what is still in doubt, when this coincidence exists, the labels can yield different results. This type of disagreement, sometimes discrepant, due to subjectivities in cataloging, led Santos and Maurer (2020, p. 12) to state that "the transparency of the methods has the potential to bring greater contributions to the public than the actual classification of information".

## 3. Methodology

Brazilian fact-checking agencies have stood out in the period of the new coronavirus pandemic, of which the main symptoms are fever, cough and respiratory distress. This pandemic began in December 2019 in China. While the agency Fato ou Fake organized all productions about COVID-19 (coronavirus disease 2019) at a new address (https://g1.globo.com/fato-ou-fake/coronavirus/) (accessed on: 7 September 2020), with a highlight on the main page and on your first call, Lupa agency, on the the other hand, allows you to browse through the verification of this information by four categories: #coronavirus, #novo-coronavirus, #COVID and #COVID-19. These two agencies were chosen to be analyzed because they represent two distinct categories: traditional media that propose to carry out the verification, even with journalistic vehicles (Fato or Fake, from the G1 portal), and media originating from digital that have been proposed to perform the same activity more independently (Lupa). Although Lupa is hosted on Grupo Folha's UOL portal, the initiative did not come from this group and there is no interference in the checks, presenting itself, therefore, as an independent agency.

The general research problem to which this article is a part carries out two complementary movements, identifying: what are the fake news patterns propagated in the

disinformation process about COVID-19? And how do reporting agencies' websites assess this false information and present the results to the public? In this text, we present the exploratory results of the initial stage of this double questioning through the frequency of checks on the pandemic on the websites of the agencies Fato or Fake and Lupa.

Following the predictions of the World Health Organization (WHO) and the Ministry of Health of Brazil, the corpus included all checks on COVID-19 in Fato or Fake and in Lupa in the first nine months of 2020: January, February and March—first cases of infection and death and definition as a pandemic; April, May and June—increase and stabilization of cases; and July, August and September—stabilization and decrease in contamination—of what was later known as the first wave. Even if this prognosis has not been fully carried out, in terms of communication of misinformation, these months were enough because they go beyond the initial period, allowing for greater circulation of information about the pandemic, not just in the period when it was new.

This research used the framing methodology, which has been increasingly explored by studies on the media (Reese 2001; D'Angelo 2002; Matthes and Kohring 2008). In the Brazilian context, Soares (2009) states that the framework receives increasing attention, especially in studies on journalistic coverage of elections and social movements. For the development of this research, it was used the notion of Entman (1993), for whom to frame is to select aspects of reality from the definition of the problem, the causal interpretation, the moral evaluation and the treatment recommendation.

After an analysis of the frequency and relevance of checks on COVID-19 at the Fato or Fake and Lupa agencies (Martins and Teixeira 2020), it was identified the thematic subtopics associated with the definition of the framework and the labels used to frame each checked fact. The verification of the dimension that these stamps bring to the checks is consistent with adaptations of the framing methodology, proposed by Vimieiro and Dantas (2009) and Vimieiro and Maia (2011), who considered, in the case of printed newspapers, the headings (or outriggers).

The evaluation of checks at the two agencies allowed for segmentation into 15 mutually exclusionary categories. Although fake news could address more than one of the topics listed below, once the spread of lies can add more than one topic, they were selected in the segment that was considered the main one, that is, the primary target of misinformation. As we specifically deal with fake news checks about COVID-19, other productions by agencies were disregarded, even if they were about the pandemic's misinformation, such as balance sheets of publications, podcasts and interviews, among others. The themes were defined by the authors, after reading all the selected checks, framing related subjects; in this way, to facilitate understanding and investigation, some categories were grouped together with more than one subject, while others, due to the high flow, were segmented.

The 15 themes are: (1) Agglomeration, about bringing people and crowds together; (2) Attendance, from the logistics of spaces and health professionals to the patient's recovery and the end of the pandemic; (3) Benefit, on possible pandemic, financial or other gains; (4) China, with xenophobia and explicit ideological bias against this country; (5) Contravention on crimes and contraventions, committed or amnesty during and related to the pandemic; (6) Distancing, unlike Agglomeration, the focus here is on fake news about isolation, lockdown and closing of establishments; (7) Disease, from the origin, contamination, diagnosis of the new coronavirus to its symptom, transmission and sequelae; (8) Examination of COVID-19 tests, from performance to results; (9) Death, from data to burials and victims' stories; (10) Personality, on individuals or legal entities, as long as they are not political; (11) Politics, from election and demonstration to specific politicians; (12) Prevention, about something that prevents contamination, such as alcohol gel, facemask, hand washing and a thermometer; (13) Prediction, alleged predictions about the disease or pandemic made in the past, near or far; (14) Medicine, about cures in general, after getting it; (15) Vaccine, from immunization to effects.

#### 4. Fato ou Fake

From January to September, Fato ou Fake made 336 publications in the section destined to COVID-19, of which only 4 were not about checks, but about balance matters. In fact, the first publication, the only one in January, was a presentation, not a proper evaluation. The other three balance sheet posts took place on April 1st, with the purpose of presenting what was true amidst so much information, and when the editorial reached 150 and 300 checks, in late April and early August.

Grupo Globo's checking agency brings only two eponymous labels to the site Fato or Fake. Of the 332 evaluations, only one confirmed the information; that is, there was only one Fact, in the Service category, about the role of the Brazilian army in helping victims of COVID-19, in the month of July, after 271 checks for lies. In total, there were 331 fake news articles in 9 months, for an average of more than one fake news article disproved per day. The Politics and Death categories were the most recurrent (Figure 1), with 41 and 41 checks, followed by Medicine and Prevention, with 38 and 37, comprising practically half (47.29%) of the 332 evaluations.

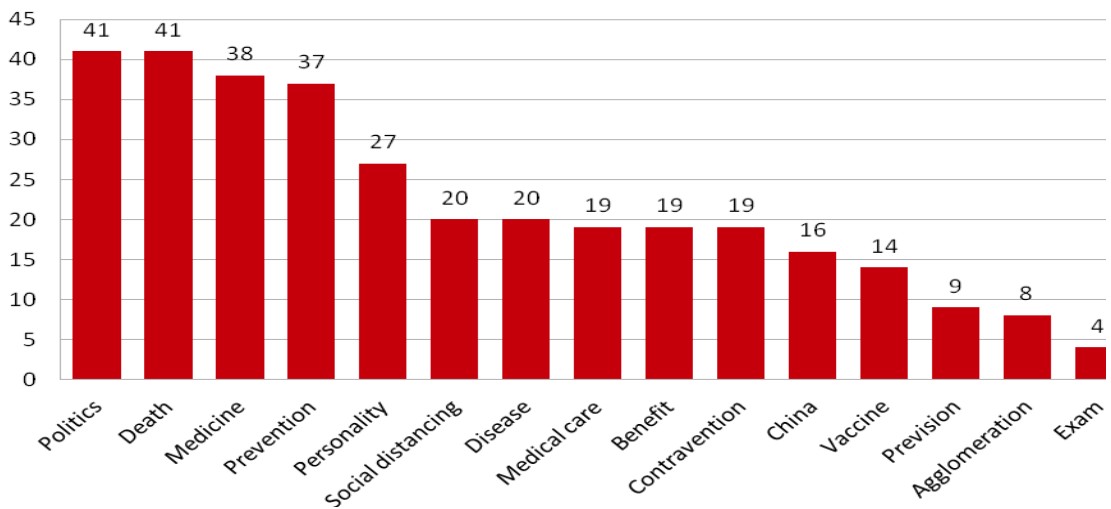

**Figure 1.** Frequency of themes in Fato ou Fake. Source: elaborated by the authors based on raw numbers.

The target of the Political category, with a total of 41 checks, was the governor of São Paulo, João Doria, with 10 appearances (24.39%). Although it is also mentioned once in Personality, here the lies were related to the China, Vaccine or some Contravention, but mainly to the Agglomeration. As happened in other categories, sometimes fake news addressed more than one theme. The lies directed at 13 governors and 3 mayors (39.02%), for example, had a link with Benefit, Distancing and Death. Russia appears twice; the first time it was said that the government had released lions into the streets to ensure isolation, and the second time that Putin's daughter had died after taking the local vaccine.

The other most present category is Death, with 41 checks, of which 25 (60.98%) are about wrong numbers of deaths by COVID-19. Additionally, in this sub-theme, three were about empty coffins and two about people buried alive. There are also two lies about people who would have died after getting the vaccine. The lies about Death brought links with Politics, as in: "It's #FAKE that deaths by COVID-19 in Amazonas decreased significantly after a visit by the Minister of Health". For this one, Politics was placed because the background (Death) is a lie, but the intention is to improve the minister's image, evident with the expression "it was only Bolsonaro sent the minister". "It's #FAKE that the government of SP reviewed the confirmed deaths of COVID-19 and more than half had a negative result" and "It's #FAKE that the Ministry of Health audits the numbers of cases and deaths of COVID-19", which could being Political, they had no association with

the mayor or governor of São Paulo, in the first, and, in the second, the biggest lie was the number of deaths.

The 38 checks on the Medicine theme focus mainly on the cure by chloroquine; although the medicine only appears from May onwards, it is in 10 evaluations (26.32%), in this category, and in 2 in Personality, involving Camila Pitanga and Ziraldo. Many homemade and natural solutions appear as a solution to get rid of the disease, but those based on lemon are in 5 checks. A lie was related to Care; however, it derives from the main fake news about the cure: "It's #FAKE that Natal zeroed patients with COVID-19 hospitalized in ICUs thanks to a medication protocol". Two checks, which could also be Prevention, were selected for Medicine because they conveyed the idea of a cure, as we defined this category: "It's #FAKE that chinese bark tea is effective against coronavirus" and "It's #FAKE that chlorine dioxide prevents and cures COVID-19".

The Prevention category had 37 checks. In addition to the mask, with 15 recurrences (40.54%), the main target of fake news on this theme, alcohol gel was mentioned 6 times (16.22%) in this theme, and once more in Politics, whose focus was to reach the governor of Pernambuco. Infrared thermometer was mentioned on two occasions. It is worth mentioning that lies involving masks appeared 17 more times, spread over several themes: China (4), Contravention (3), Personality (2) and Politics (5). Prevention brought lies about how to avoid contamination not only related to the Medicine, as in the two previous examples, but also to Contravention, as in: "It's #FAKE that a driver who drives without a coronavirus protection mask can lose points in his driver's license", in that the lie tries to target the use of masks and not the points on the driver's license.

Personality normally brings another embedded theme, although the emphasis is on the socially relevant person (natural or legal). This happened in 26 checks, of which 20 (76.92%) were published between March and May. In five (19.23%), they mention the WHO (World Health Organization) to demoralize them, mainly associating them with Contravention or Distance. Bill Gates appeared three times (11.54%), in one of them linked to the Vaccine, although the focus is on the entrepreneur: "It's #FAKE that Bill Gates finances non-liquid vaccine against COVID-19 that installs chips in people". "It's #FAKE that the video shows actor Fabio Assunção criticizing Doria and the measures of social isolation" which, although it is related to Politics and Distance, the focus is on the actor who works at Grupo Globo, the same company mentioned in other fake news: Caco Barcellos, Camila Pitanga, Maria Júlia Coutinho, Renata Vasconcellos and Thiago Leifert. The Catholic religious Fábio de Melo and Pope Francis are mentioned, with the Argentine appearing twice. Footballers Messi and Cristiano Ronaldo also appear, in addition to singer Bell Marques and intellectual Mario Sergio Cortella.

The Distancing theme appeared 20 times, with the lie about the closing of establishments gaining a lot of attention, being in 5 (25%). In "It's #FAKE NY Governor said survey by the state Department of Health proved ineffective social isolation", while there is a link to Politics, the lie's emphasis is on Distancing. Disinformation brings untruths about the policies adopted and their consequences in cities in the South and Southeast regions, such as Caxias do Sul, Curitiba, Rio de Janeiro, São Paulo and Pelotas, and in some countries, such as South Korea, Israel and the Netherlands. To support a defense against social isolation, there were also 3 fake news articles with alleged research from Harvard and USP universities and a British neuroscientist.

In the 8 Agglomeration checks, the opposite of the previous theme, 2 were about public places in Rio de Janeiro, such as the beach and funk parties, 2 about a popular restaurant and the subway in São Paulo, and 1 supposed agglomeration at the bank Caixa Econômica Federal to receive emergency aid. All these lies are about circulations in spaces aimed at the poorest population, with none about spaces destined for the higher classes. Another two are about agglomerations in Germany. There was a link of this category with Benefit, Contravention and Prevention, as in "It's #FAKE that a photo shows the bank agency of Caixa Economica Federal full after an emergency aid sanction of R$600" and in "It's #FAKE that a video shows funk parties and no one with a mask in favela in Rio after the decision

of Supreme Courte from Brazil to prohibit police operations", although the main one was in the gathering of people in the same place.

As it was an unknown virus, the Disease itself was in 20 checks, 15 (75%) showing up until May, when many uncertainties still circulated. Among the main sub-themes, three (15%) lies spoke of the origin and five (25%) of the dissemination and transmission of the disease. Attendance had 19 evaluations, with 17 (89.47%) only from April to June, dealing with lies about logistics and health professionals until the patient's recovery and the end of the pandemic. The main issue in this category, however, was about the non-occupation of spaces, with 13 (68.42%) recurrences, as if the patients were cured and the country was on the right path in combating COVID-19.

The Benefit category also had 19 checks, with 17 (89.47%) only from March to May. Although the focus was on the financial aspect, especially on the aid that the government would give to the population, such as Bolsa Família and, mainly, the emergency aid, in seven (36.84%), there were broader and generic gains, such as: "It's #FAKE that flamingos occupied the Venice canal amidst the coronavirus pandemic". In other cases, the lie was related to other themes, such as Contravention and Prevention, in the first, and Service, Death and Politics, in the second case: "It's #FAKE that video shows prisoners without masks in line to receive assistance at a bank in the middle of the pandemic", in which the greatest consternation was in the prisoners being, supposedly, benefited; and "It's #FAKE that the Ministry of Health transfers R\$12,000 to hospitals for each death by COVID-19".

Contravention, a more tangential issue, appeared in 19 assessments, with 17 (89.47%) from March to June and 7 (36.84%) related to Distancing, although focusing on police action and approach, abroad or in Brazil. The plundering of business establishments was in 2 checks. Another lie was associated with Vaccine, however, with an emphasis more on the oversight power of the State: "It's #FAKE that the government cannot force people to be vaccinated against COVID-19".

Although without as much specific checking as other themes mentioned above, with 16, China is certainly one of the most prominent, especially if we consider that the country was often used as a kind of aggravating factor for the other lies evaluated. Of the total, 14 (87.50%) of these assessments took place in the first months of the pandemic, from February to April; 5 (31.25%) dealt with contaminated products coming from China and 3 of these specifically on imported masks. This connection with the Asian country also brought up themes such as Contravention and Death, although the lie focused on China or Chinese, as in: "It's #FAKE that video shows a driver arrested by SWAT in China for having coronavirus", "It's #FAKE that video shows Chinese woman with coronavirus being arrested in a market in Australia after spitting on bananas" and "It's #FAKE that photo shows hundreds of dead in China." Besides these, in "It's #FAKE photo that shows Rodrigo Maia hugging Chinese President Xi Jinping", the Political category was not chosen, as the emphasis is on the President of China, not Maia—that is, more because of the government system of that country.

The Vaccine, with 14 evaluations, had more than half of the lies about it, 8 (57.14%), appearing as of July, that is, as studies on COVID-19 advanced. This theme was closely related to China and CoronaVac, in four (28.57%) cases. Although there is an association with the Asian country, mainly in the expression "Chinese vaccine", the focus of the lie was more centered on the immune system. The COVID-19 pandemic preview was in 9 checks, and was even quoted by Nostradamus, the supposed 16th century seer. Additionally, popularly known for supposedly predicting real life situations, the animated series *The Simpsons* appears in one denied article of disinformation. Besides them, three books, two magazines, one painting and one movie are mentioned. Finally, Exam was the topic with less frequency, with only four, and not specifically dealing with false positives or negatives, but aspects such as exams at home, the speed of the result, test sequelae or purchases made years before the start of the pandemic.

### 5. Lupa

Specifically about the posts by the Lupa agency, it is important to point out that the articles linked to special projects such as "Lupa na Ciência", "Podcast Verifies Coronavirus", "Caiu na Rede" and "Latamchequea Coronavirus" were excluded. These posts are not really checks, but broader journalistic texts or audios discussing COVID-19, so much so that the content posted in Spanish, in general, are just repetitions of previous checks translated. Thus, of the 522 publications on the pandemic, 324 were, in fact, checks.

To determine the veracity or not of the content of a check, the agency proposes, according to the labels displayed on the right side of its main page, to classify the contents into nine categories: (1) "True", when the information is proven to be correct; (2) "True, but . . . ", in cases where the information is correct, but the journalist believes that the reader deserves further explanation; (3) "It's still too early to say", when the information may turn out to be true, but it still is not; (4) "Exaggerated", in situations where there was an exaggeration of information, even if it is on the right path; (5) "Contradictory", when the sources presented in the content contradict each other; (6) "Underestimated", in cases where the data is more serious than the information that was presented; (7) "Unsustainable", when there is no public data to confirm it; (8) "False", for information that is proven to be incorrect and (9) "Keeping an eye", in situations where the content will remain being monitored.

During the analyzed period, the False label was massively used to classify the checks on COVID-19; there were 320 (98.77%) of posts in this sense, not being present in only four occasions. In the articles that determine posts circulated on various social networks or in different formats, some were classified with two or more labels, but always dividing the classification with the idea that the information is false, with this specific label. Among the labels that were applied alone, and not matching False, were found: True (2 cases, in the Disease and Prevention categories); Unsustainable (1 case, in Disease); and Exaggerated (1 case, in Medicine). Regarding the themes (Figure 2), it is possible to state that Politics and Death were the ones that most appeared, with 48 and 39, followed by Medicine, Personality, Prevention and Care, with 32, 30, 29 and 28 cases, respectively, comprising more than half (63.58%) of the 324 evaluations. Agglomeration and Exame, on the other hand, point to much smaller numbers, 6 and 3 checks, and Prediction with two, with the other categories being between 23 and 15 assessments.

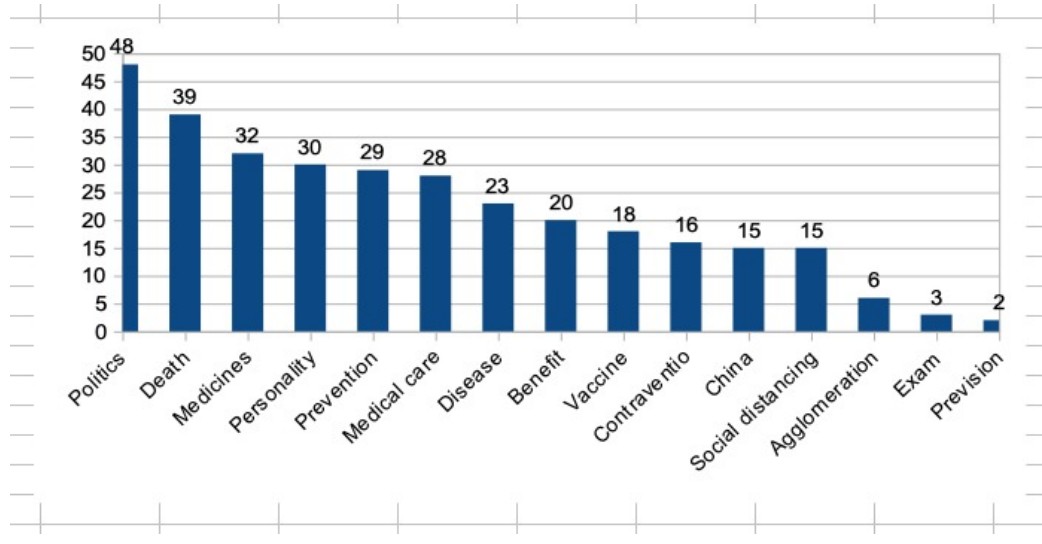

**Figure 2.** Frequency of themes in Lupa Source: elaborated by the authors based on raw numbers.

The Political theme had the most expressive percentage of checks: 48, we included any political issue that encompassed the most direct actions of governments or the figure of their politicians. The attacks on the current governor of São Paulo, João Doria, were

identified in 13 checks, of which 11 (22.92%) are in this category, and the 2 appearances are in the themes China and Medicine. In "It is false that Doria banned chloroquine in hospitals in São Paulo" and "It is false that Doria took 'Chinese vaccine' against COVID-19 without wearing a mask", the lies are articulated, for example, with Remedies and Care, in the first case, and China and Prevention, in the second; however, the focus of the lie is Doria. Still, in Politics, "It is false that the government of Paraíba used emergency funds to hire artists" and "It is false that the number of deaths per day fell in Manaus after Teich's visit" bring associations with Benefit and Death; however, the intention is to attack mainly the governor, on one side, and improve the image of the minister and president, on the other.

Death, totaling 39, has 8 (20.51%) false allegations that living people were being buried or that coffins were being buried empty; in addition to these cases, we highlight three fake news articles about business suicides. These lies sought, above all, to reinforce the false dichotomy between health and economy, raised by some politicians and businessmen during the pandemic.

In Medicines, 32 cases were identified, 27 (84.38%) of them were dedicated to exalting the benefits of chloroquine, in which research already demonstrated its ineffectiveness—that is, more than not having enough information, as happened during our sample, since the end of 2020, its use was already unauthorized. This theme presents relationships with the following categories: Politics—with associations to the governors of Bahia, Rio Grande do Sul and São Paulo, in addition to the mayor of that city, minister Augusto Heleno and the PT (political group) and STF (Brazilian supreme court); Personality, citing an actress who would have been cured with chloroquine and Prevention, with a lie that presents a formula to prevent and cure at the same time. Chloroquine still appears in the categories of Politics, mentioned above, linked to the governor of São Paulo, João Doria and Personality being associated with cartoonist Ziraldo, mentioned below, shows that fake news is always seeking to acquire some notoriety from socially recognized people.

Personality totaled 30 cases, mainly with people linked to political agendas, from businessmen (Luciano Hang and Roberto Justus) to artists (Camila Pitanga, Eduardo Costa, Preta Gil, Letícia Spiller, Ziraldo) and TV Globo presenters and journalists, such as Fátima Bernardes, Maju Coutinho, Renata Vasconcellos (2), Tiago Leifert and William Bonner—Luciano Huck was mentioned in the Prevention category, presented below. Eleven (36.67%) lies involved large companies, considered here as legal personalities, such as Atacadão, Brahma, Carrefour, Extra, Lojas Americanas, Renner, Samsung and Xiaomi, saying that they would do some improvement (Benefit) or that they did not comply with any standard (Contravention). The figure of Bill Gates appeared 3 (10%) times, and always related to the Vaccine.

Attacks through false information about coronavirus prevention measures, which do not refer to Agglomeration and Distance, which were later treated isolated, were grouped in the Prevention category, which included 29 cases. The most attacked measures were masks and alcohol gel, appearing in 10 (34.48%) and 6 (20.69%) checks. In this category, there were also lies about infrared thermometers and other misinformation even quoted Luciano Huck, relating to Personality, but with a focus on prevention, about a product that would improve immunity.

In Attendance, there were 28 cases of fake news, of which 13 (46.43%) said that the hospitals would not be crowded or in a critical situation. In this theme, we identified the attempt to show positive actions of certain institutions, without these being actually carried out. It is also possible to observe that 26 (92.86%) checks in this category appeared until June, in the first six months of 2020. In Disease, 23 checks related to the new coronavirus were grouped, whose fake news mostly concern 14 (60.87%), to the inexistence of COVID-19, based on lying, old or inaccurate data. One said that the name of the disease was engraved on a train and another was related to Prevention, although the focus was on its effects: "Post on Facebook distorts information about the immune system, alcohol gel and

COVID-19". Seventeen (73.91%) lies were checked in the first six months of the year, when the pandemic was new.

In Benefits, 20 cases were identified, with an emphasis on three months since, from March to June, there were 16 (80%) checks. In this category, some lies were recurrent, such as resources received for each death of a COVID-19 patient, related to Death, and suspension or receipt of emergency aid, linked to the Politics. These two sub-themes had 3 (15%) ratings each. Still, the main targets in Benefit were scams and fraud, when people signed up to supposedly receive a benefit; although they are associated with Contravention, the 8 (40%) checks had a possible gain as a primary lie. Distancing, China, Disease, Personality and Medicine were other themes that were related: "It is false that elderly people who disrespect the isolation of COVID-19 will have their retirement suspended", "It is false that an American researcher was arrested for manufacturing and selling the new coronavirus to China" and "Madagascar's President did not say WHO offered bribes to poison 'COVID-19 cure'".

The vaccine appeared in 18 checks, intensifying after the peak of the pandemic in the country. Even so, speculations already existed, with four appearing before June—all in March—and 14 (77.78%) from June to September, really demonstrating that the subject gains more prominence after the consolidation of the pandemic, although the subject is found in themes such as China and Politics because of the image of the Asian country in Brazil and the performance of the governor of São Paulo, João Doria; in the Vaccine category, the relationship with the state of São Paulo occurs twice and with CoronaVaC, or "Chinese vaccine" on 6 (33.33%) occasions. The verification of fake news related to changes in human DNA from the application of the vaccine is also recurrent, in 3 (16.67%) checks. Doubts in this regard are even based on preliminary tests that would have been carried out in countries such as Cuba, Israel, France and Australia, with 4 (22.22%) evaluations.

Sixteen cases of checks on the Contravention theme were identified, with examples ranging from the association of crimes with drug users, with two specifically on Cracolândia, to aggression in police approaches and commercial establishments that would be looted during the quarantine. Each of these sub-themes brought 3 (18.75%) checks each. The category appears only from March to July 2020, the height of the period of protective measures by state and municipal governments across the country, revealing that fake news was aimed at generating fear in the population also with violence, in addition to the disease. It is also possible to present link with the themes Benefit and Prevention, in: "It is false that the federal government dismissed 396,000 public servants who defrauded emergency aid" and "It is false that a bus driver was attacked with a cobblestone for complaining about a passenger without mask".

The China category features 15 checks. In these cases, we do not point out the relationships established with the Vaccine just for bringing the expression "Chinese", nor the supposed Benefit obtained by the Chinese government from the coronavirus. Even so, it was possible to notice a fake news bringing the subject of the Vaccine, in "Volunteer who reported fever and body aches did not take Chinese vaccine, but Oxford vaccine"; although it is related to the immunizing agent, the focus is on China, as the vaccine that gave a reaction was the one from Oxford, although they soon blamed the Chinese. There was also one involved with Politics: "It is a false tweet by deputy Kim Kataguiri about a visit to the Chinese president after a pandemic", in which China serves to "worse" the representative's image.

Two categories have a direct link: Distance and Agglomeration. In the first, there were 15 assessments, of which six (40%), instead of encouraging the protocol to combat COVID-19, make clear the intention to question social distancing, citing the WHO, in 3 (20%) checks, and foreign countries, in four (26.67%). The 6 fake news articles on Agglomeration, in turn, leave doubts about their intention, as it is difficult to specify whether they seek to denounce irregular agglomerations or discourage those who were complying with isolation protocols. This complication is noticeable in "Photo of a crowded wagon in SP is old and has no link with the expanded rotation in the pandemic", in which the information, duly checked and

labeled as False, conveys the idea that the rotation in São Paulo generated more problems than solutions to fight the new coronavirus.

The Exam carried out to identify who is or has been with COVID-19 was pointed out by scientists and authorities as one of the ways to fight the pandemic. For this reason, mass testing was greatly encouraged in several countries, and three evaluated lies attack these tests and generate fear in the population that the test affects the brain, the result comes out automatically and criminals distribute it. In Prediction, we identified 2 cases, one on The Simpsons—a series known as precognitive on several subjects—on 3 March 2020, a week before the WHO declared the disease a pandemic, and another in June, involving a book of the 20th Century by C.S. Lewis.

## 6. Final Considerations

Checking agencies began to devote more attention to evaluations of COVID-19 fake news. As the pandemic intensified, the frequency of analyzed lies increased, reaching, from January to September 2020, 332 in Fato ou Fake and 324 in Lupa, which were segmented into 15 mutually exclusive themes. Although each check was in only one category, fake news typically addresses more than one lie, yet each has a primary target.

The importance that the vehicles gave to the verification of really false information, considering that only one was true in Fato or Fake and two had this label in Lupa, demonstrates that the interest is not in confirming the information—allocating the function to journalism to inform correctly—but to the correction and explanation of the lies. There are those who still think of agencies "as accessory lines of traditional journalism, as they constitute channels for legitimizing the content produced by the vehicles" (Fernandes et al. 2019, pp. 19–20). This perspective, however, does not prove to be completely true, as the main Brazilian fact-checking agencies are not part of the mainstream media and intend to carry out work that is different from traditional journalism, not basing themselves or needing it for their existence.

Of the subjects most evaluated by the two agencies, Politics and Death were the two main ones, followed by Medicine, Prevention and Personality. The high frequency of Politics reveals the ideological polarization that Brazil is experiencing, leading to a global health crisis such as the pandemic of the new coronavirus. It is no coincidence that one of the recurring figures in fake news with more than 20% was João Doria, governor of São Paulo who supported President Jair Bolsonaro in the election and is now an opponent.

The themes Death, Medicine and Prevention refer specifically to the heart of the disease, as its major consequence, death, as well as the possibility of getting rid of it, either preventively or after contracting it. In the first case, the lies were around the falsifying number of deaths, especially with false burials and empty coffins, reaching more than 60% in Fato or Fake. As for the cure, chloroquine prevailed, demonstrating even more the political presence in the lies, reaching almost 85% of the checks in Lupa, in the Medicine category. Finally, lies about mask and alcohol gel dominated the last theme, with practically 55% of recurrence in both agencies.

The high verification number in Personality demonstrates the importance that some public figures have in our society, especially as a demand for them to have exemplary behavior. This relevance is also perceived in fake news from people or works that would have made the COVID-19 Pandemic Prediction. Even though he is not Brazilian, Bill Gates, the founder of Microsoft, appears in at least 10% of the lies on this topic. Another figure that stood out as a target of misinformation was the World Health Organization, although it was not always in this category.

The most common of the lies was the correlation between the subjects, being sometimes difficult to define in just one theme. It was common to observe the words "Chinese" as a negative variant for fake news in the Political and Vaccine categories, always with the intention of disqualifying because of a possible origin from the Asian country, a discourse that mixes ideological prejudice with xenophobia. Still, alone, China appeared in 16 checks on Fato or Fake and 14 on Lupa.

Distancing has always appeared more than Agglomeration, although Distancing is greater in Fato or Fake, demonstrating that the lie was more focused on a possible solution to the pandemic than on possible damage caused by these actions. Even so, in the first one, there was an emphasis on closing establishments, with an attempt to spread misinformation about actions that would affect the economy and people's lives. Service, Benefit, Contravention, Disease and Vaccination were also relevantly contained in the fake news analyzed by the two agencies, although Benefit and Contravention are presented as more transversal issues, related, above all, to emergency aid provided by governments, i.e., to the supposed benefits that they led to coups and police action to ensure distance and avoid agglomeration

The Attendance, in turn, brought misinformation about the effectiveness of the treatment against the new coronavirus, above all to lie that Brazil was achieving good results, with implicit ideological interests; were almost 69% in Fato or Fake and 47% in Lupa. Disease, even if it treated from the origin to the end of COVID-19, did not stand out in relation to these more parallel issues, as did the Vaccine, which acquired more relevance in the final third of our corpus, and usually related to China, around 30% in both agencies. Agglomeration, Exam and Prediction were the categories with the lowest recurrence, with no major impact. In later works, other aspects of the checks should be highlighted in order to have a holistic view of this disinformation process, such as the origin and formats of fake news and, above all, the media resources used to better disprove them, also emphasizing the structure of the title and the recurred sources.

**Author Contributions:** Conceptualization, J.T. and A.M.; methodology, J.T.; software, A.M.; validation, J.T. and A.M.; formal analysis, J.T. and A.M.; investigation, A.M.; resources, A.M.; data curation, A.M.; writing—original draft preparation, J.T. and A.M.; writing—review and editing, J.T.; visualization, J.T.; supervision, J.T. and A.M.; project administration, J.T. and A.M.; funding acquisition, J.T. and A.M. All authors have read and agreed to the published version of the manuscript.

**Funding:** This research received no external funding.

**Informed Consent Statement:** Not applicable.

**Conflicts of Interest:** The authors declare no conflict of interest.

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
