# Peer review of "Thematic Patterns of Disinformation about COVID-19: The Framing of Checks in the Fato ou Fake and Lupa Agencies"

_journalmedia, doi:10.3390/journalmedia3010003_

Round 1
Reviewer 1 Report
- The study is very relevant, current and with great scientific relevance.
- The methodology is consistent with the proposal of the study: analysis of the action of two Brazilian fact-checking agencies, taking into account the accelerated volume of misinformation and disinformation that the Covid-19 pandemic accelerated and increased around the world.
- The sample is very significant and the analysis technique very suitable for the study objectives.
- Necessary improvements:
- Expand and diversify the theoretical framework;
- Review the formal norms indicated by Media and Journalism;
- Graphical data should be translated into English.
Author Response
Thank you for your comments. We included your suggestions in the manuscript. See the attachment.

Reviewer 2 Report
I thought this was an interesting study. Here are some comments that I hope are helpful toward revision.
The literature review could be strengthened by referencing the emerging literature on “dark participation,” which helpfully moves the field away from the technologically deterministic and utopian literature on the “democratization” of journalism through things like participatory journalism and user-generated content. Key sources here include De Vreese (2021), Frischlich et al. (2019), Quandt (2018), and Westlund (2021).
I would like to see a clearer definition of fake news in the front end of the paper. A good start for this might be Tandoc (2019), Tandoc et al (2018), and/or Tandoc et al (2021).
I think it would be helpful if, at the end of the literature review, the author(s) posed clear, empirical research questions to indicate what the focus of the analysis is.
I couldn't discern from the method section (which ought to have its own heading and be designated as such) how many articles comprised the sample for each of the news organizations studied. Can this be clarified?
Is the Graph on p. 4 reporting raw numbers or percentages? I assumed raw numbers but it wasn’t immediately clear to me.
In terms of writing and organization, it would be helpful if there were more subheadings to organize the narrative more clearly.
The discussion/conclusion would benefit from a section more clearly focused on what explains the differences that were found (i.e., what inferences can you make from your findings). This would nudge the paper slightly beyond description into explanation.
References for sourced cited above
De Vreese, C. (2021). Beyond the darkness: Research on participation in online media and discourse. Media & Communication, 9(1), 215–216. https://doi.org/10.17645/mac.v9i1.3815
Frischlich, L., Boberg, S., & Quandt, T. (2019). Comment sections as targets of dark participation? Journalists’ evaluation and moderation of deviant user comments. Journalism Studies, 20(14), 2014–2033. https://doi.org/10.1080/1461670X.2018.1556320
Quandt, T. (2018). Dark participation. Media & Communication, 6(4), 36–48. https://doi.org/10.17645/mac.v6i4.1519
Tandoc, E. C. (2019). The facts of fake news: A research review. Sociology Compass, 13(9), 1–9. https://doi.org/10.1111/soc4.12724
Tandoc, E. C., Lim, Z. W., & Ling, R. (2018). Defining “fake news”: A typology of scholarly definitions. Digital Journalism, 6(2), 137–153. https://doi.org/10.1080/21670811.2017.1360143
Tandoc, E. C., Thomas, R. J., & Bishop, L. (2021). What is (fake) news? Analyzing news values (and more) in fake stories. Media & Communication, 9(1), 110–119. https://doi.org/10.17645/mac.v9i1.3331
Westlund, O. (2021). Advancing research into dark participation. Media & Communication, 9(1), 209–214. https://doi.org/10.17645/mac.v9i1.1770
Author Response

(The authors gave the same response as above.)

Round 2
Reviewer 2 Report
Thank you for the diligent work on this revision!